# Design of advanced airfoil for stall-regulated wind turbines

**Francesco Grasso[1], Domenico Coiro[2], Nadia Bizzarrini[2], Giuseppe Calise[2]**

[1] Aerodynamix, Napoli, 80128, ITA – Contact Author: skyflash@inwind.it

[2] Dip. Ingegneria Industriale, Università di Napoli FedericoII, Napoli, 80123, ITA

**Abstract.** Nowadays, all the modern MW-class wind turbines make use of pitch control to optimize the rotor performance and control the turbine. However, for kW-range machines, stall-regulated solutions are still attractive and largely used for their simplicity and robustness. In the design phase, the aerodynamics plays a crucial role, especially concerning the selection/design of the necessary airfoils. This is because the airfoil performance should guarantee high wind turbine performance, but also the needed machine control capabilities. In the present work, the design of a new airfoil dedicated for stall machines is discussed. The design strategy makes use of numerical optimization scheme where a gradient-based algorithm is coupled with the RFOIL code and an original Bezier-curves-based parameterization to describe the airfoil shape. The performances of the new airfoil are compared in free and fixed transition conditions. In addition, the performance of the rotor is analysed comparing the impact of the new geometry with alternative candidates. The results show that the new airfoil offers better performance and control than existing candidates do.

## 1. Introduction

Looking back in wind turbines history, pitch-regulated machines gradually substituted stall-regulated systems. In fact, the possibility to optimize the power production for each wind condition by regulating the pitch angle of the blade, proved to be a key feature to maximize the Annual Energy Production (AEP) of the wind turbines. Nowadays, all the modern MW-class wind turbines are "by default" pitch-regulated and several innovations are implemented by Industry to improve the pitch performance (e.g. individual pitch control, fine regulation mechanisms/algorithms) and extract more power.

In apparent contradiction with MW machines however, small and medium kW wind turbines are still largely stall-regulated machines. The reasons of this are easy to explain. In fact, the advantages of the pitch system come with some costs. The first is the direct cost of the pitch system and its maintenance. Secondly, the pitch system increases the general complexity of the system, together with the development costs and the issues related to the system robustness/reliability. Extra components, such as onboard anemometers and pitch bearings are necessary to operate the pitch of the blade correctly. All these costs and complications can be very relevant for small machines and it explains why a robust and easy-to-maintain solution is preferred even with some AEP sacrifice.

From the design point of view, the stall-regulated machines still offer a challenging task, especially concerning the aerodynamics of the blade that should ensure the power performance but provide the machine control. In practice, the design of the blade should obviously aim to maximize the AEP, but it is also the only component to keep the turbine under control, stopping it when necessary. To do so, the stall and post-stall characteristics of the airfoils play a crucial role. From this angle, the selection/design of the airfoils and the blade shape design are more delicate than pitch-regulated turbines.

The present work focuses on the design of a new airfoil specifically designed for stall-regulated
turbines. The next section illustrates the design of the new airfoil in comparison with existing
geometries. Then, its impact on the overall turbine performance is discussed.
**2. Design of the new airfoil**
*2.1. General requirements*
The selection of the proper airfoils is very relevant to achieve satisfactory wind turbine performance.
Depending on the area of the blade, the requirements change quite a lot; in fact, the outer sections are
optimized for high aerodynamic performance, while the inner sections are designed to provide low-
weight and structural integrity to the blade.
The focus of the present investigation is the outer region of the blade, so the airfoils should have high
aerodynamic efficiency (L/D). This is the primary parameter to increase the annual energy production
of the rotor, but it is not the only one. Besides that, the stall behaviour should be considered, avoiding
sharp stall. This would lead in fact to load problems to the blade (e.g. fatigue issues and additional noise)
and other components. The impact of roughness on the rotor performance should be also addressed when
the airfoil is designed/selected. Normally, the annual production decreases when the blade is
contaminated by dirtiness (e.g. mosquitos), damages (e.g. erosion) or imperfections. Designing an airfoil
that is robust (or less sensitive) to roughness would contribute to maintain a stable performance on the
long run. Thus, it is important to have airfoils with reduced drop in maximum lift coefficient and
aerodynamic efficiency in rough conditions. In addition, limited variations in terms of corresponding
angles of attack are desirable.
Looking at the blade construction, it  must be buildable and lightweight to save the production costs, so
the airfoils adopted should not have critical features which may compromise those aspects (e.g. too thin
trailing edge, very concave-complex areas). Inevitably, there is interaction between weight minimization
and annual energy production optimization, where the first would drive for instance, to large thickness
distribution to accommodate a structurally efficient spar and maximize the section's moment of inertia,
while the second would tend to reduce the airfoil thickness to reduce the drag.
A complete discussion can be found in Grasso, 2011.
*2.2. Aerofoils for stall-regulated wind turbines*
In addition to what was presented in the previous paragraph, special considerations should address the
peculiarity of stall-regulated wind turbines. As mentioned, the big challenge of these machines is their
control. While the pitch-regulated turbines can change the pitch angle of the blades, so to optimize the
performance for each wind speed, the stall-regulated turbines are much simpler and rely only on the
aerodynamics of the airfoils. This increases the complexity of the airfoil design
First of all, the airfoils of stall regulated turbines work in a quite wide range of angles of attack so a
sound performance comes from the fact that they achieve high aerodynamic efficiency over the angle
of attack range. This is an important element to properly setup the design process. In fact, a design point
close to stall would be desirable to obtain best AEP performance and the margin must be carefully
calibrated and reduced compared to the values for pitch-regulated machines. The stall mechanism stops
the turbine when the loads are becoming too large; postponing the stall could lead to excessive forces
on the blades and the other components of the turbine. Furthermore, the capability to control the
machine, slowing down the rotor and avoiding over-power issues depends on the airfoil stall and post-
stall behaviour. In fact, a slope of the lift curve excessively "flat" could be insufficient to control the
turbine (and so prevent over-power), while sharp stall would make more difficult to re-start the machine
and would cause sudden changes into the loads faced by the blades. In addition to this, the airfoil post-
stall response is fundamental to avoid stall-induced vibrations, which is one of the main issues to address
in designing stall-regulated machines.

*2.3. The stall-induced vibration phenomenon and its impact on airfoil design*
When a wind turbine blade vibrates, the aerodynamic forces have an additional component originated
by the vibration velocity. Such component with good approximation can be considered proportional to
vibration velocity, thus it actually acts as a viscous damping force, usually denoted as "aerodynamic
damping" (see Petersen et al., 1998, Rasmussen at al., 1993, Rasmussen, 1994). When the airfoils are
in stall conditions, the slope of the lift curve becomes negative and can cause a local negative
aerodynamic damping in the lift direction.
As an example, a descending airfoil will see an increasing angle of attack that will cause a lower value
of lift coefficient; this will be equivalent to have a component of the aerodynamic force promoting the
descent of the airfoil, thus acting as a negative damping force.
If global aerodynamic damping of the blade is both negative and larger (in magnitude) than the structural
damping, any disturbance can cause divergent oscillations which can dramatically increase fatigue loads
and can even lead to rapid failure in the worst case.
This phenomenon is usually reported as "stall induced vibrations" and represents a key issue for stall
regulated wind turbines, which work in stalled conditions for a significant part of the lifetime.
Stall induced vibrations have to be regarded as instabilities of the blades that can take place due to any
initial disturbance. A sharp stall leads to lower damping force and so larger vibrations. On the other
hand, a flat lift curve beyond the stall could be insufficient to control the turbine.
Low stall induced vibrations and power control represent two conflicting requirements which make the
design of a stall regulated wind turbine a highly complex challenge. Finding a good compromise
between these two aspects has been one of the main efforts in this work.
During the preliminary design phase, a simplified expression of the aerodynamic damping of the blade
has been used to predict the dynamic behaviour of the blades without the need of any aero elastic
analysis, to make the design as fast as possible.
The linearized approach presented by Petersen et al., 1998 has been applied to obtain a simplified
expression for the local aerodynamic damping on the different sections of the blades, only using quasi-
steady, 2-D aerodynamics of the airfoils. Then, a simplified modal approach has been implemented to
evaluate the aerodynamic damping of the complete blade, obtaining a damping coefficient (DC) used as
an index of eventual oscillations amplitude. The use of this damping coefficient has been validated with
several cases of wind turbines obtained during the optimization process, giving always results coherent
with the behaviour of the blades evaluated through aero elastic analysis.
From the expression of the local damping coefficient in the out-of-plane direction (that usually is very
close to the flap-wise direction), it is possible to notice that a gentle stall of the airfoils along the blade
(which means a small value of the absolute value $|\frac{dCl}{d\alpha}|$ beyond the stall) would be desirable to avoid the
occurrence of stall induced vibrations. The expression of modal damping coefficients (both in edge-wise
and in flap-wise directions) provides another useful information for the optimization process. For each
direction and for each mode, the modal aerodynamic damping coefficient can be interpreted as a linear
combination of the local damping coefficients of the different sections along the blade, each one
multiplied by the local displacement related to the mode shape. Looking at typical modes shapes of a
wind turbine blade, considered as a cantilevered beam, it can be observed that the highest displacements
always occur on the outer part of the blade. This means that the largest contribution to the damping of
the blade is given by the outer sections. Thus, the blade optimization to avoid stall-induced vibrations
can be limited at this part of the blade.
Typical effect of using an airfoil with a smoother stall in the outer half of the blade is shown in the
following figure, in terms of power curve and modal aerodynamic damping coefficient (DC). It can be
noticed how a gentle slope of lift coefficient curve of the airfoils (Airfoil 2) results in a reduction of the
absolute value of DC with the related stall induced vibrations but in a less power control at high wind
speeds.

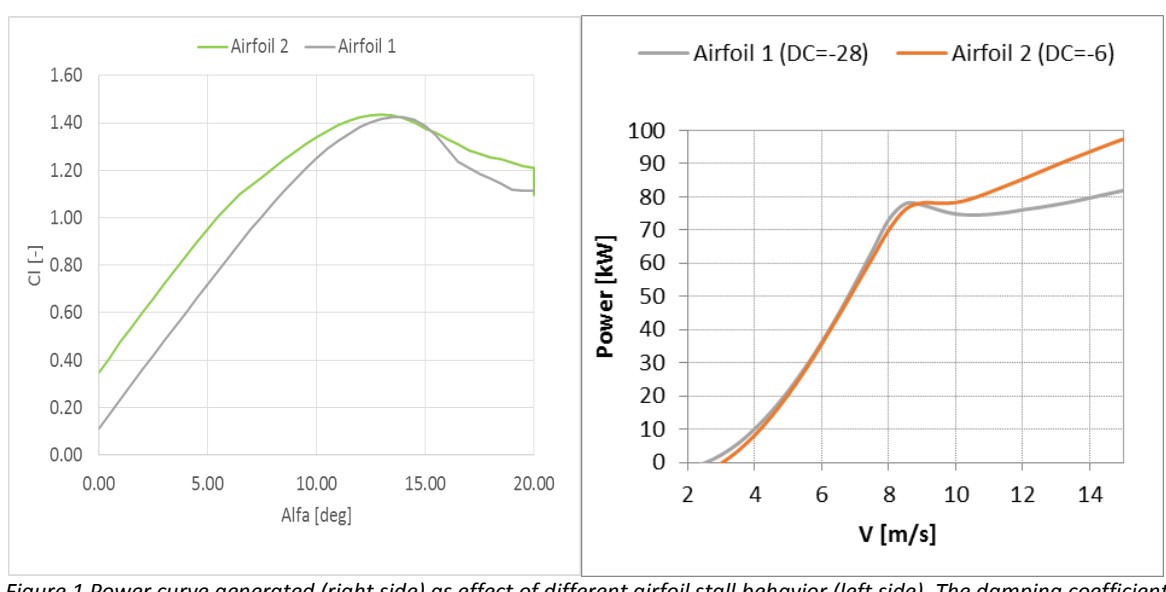

*Figure 1 Power curve generated (right side) as effect of different airfoil stall behavior (left side). The damping coefficient*
*(DC) for both cases is indicated.*
So overall, it is important that the stall margin is reduced but with gentle and continuous stall. To limit
the problem of power control the airfoils along the blade should have a low lift coefficient beyond stall
and the drag coefficient as high as possible.
To complete the challenging scenario, these characteristics must be achieved both in clean and rough
conditions. This introduces more complexity for the designer. In fact, special attention should be put in
ensuring that the characteristics of the lift curve do not change significantly in regards of stall and post-
stall behaviour.
During the rotor design, the 'rough' power curve is considered because it is the most conservative in
terms of overall performances and power control. The 'clean' power curve is considered because it is
the most conservative for extreme and fatigue loads (due to higher stall induced vibrations caused by a
more abrupt stall).
*2.4. Design methodology*
Multidisciplinary Design Optimization (MDO) (see Fletcher, 1987) has been adopted in this work. In
fact, when compared to a traditional design technique (e.g. inverse design), MDO leads to a more
accurate and computational-time saving design product, while covering constraints coming from
different disciplines. Based on author's previous experience (see Bizzarrini et al. 2011, Grasso, 2012),
a gradient-based algorithm (Zhou et al., 1999) has been preferred to control the design procedure, where
the popular tool RFOIL (van Rooij, 1996) is used to evaluate the aerodynamic performance of the airfoil.
RFOIL is a modified version of XFOIL (Drela, 1989) featuring an improved prediction around the
maximum lift coefficient and capabilities of predicting the effect of rotation on airfoil characteristics. In
fact, numerical stability improvement is obtained by using the Schlichting velocity profiles for the
turbulent boundary layer instead of the Swafford velocity profiles (Schlichting and Gersten, 2017).
Furthermore, the shear lag coefficient in Green's lag entrainment equation of the turbulent boundary-
layer model is adjusted, and the deviation from the equilibrium flow is coupled to the shape factor of
the boundary layer.
Figures 2 and 3 show a comparison between the two codes against S814 airfoil (Somers and Tangler,
1997) wind tunnel data (Somers and Tangler, 1994). As it can be observed, RFOIL accuracy for stall
region is significantly better than XFOIL and, as mentioned in the previous chapters, stall is quite crucial
parameter in this case. Additional validation tests can be found in Grasso, 2011 and van Rooij, 1996.


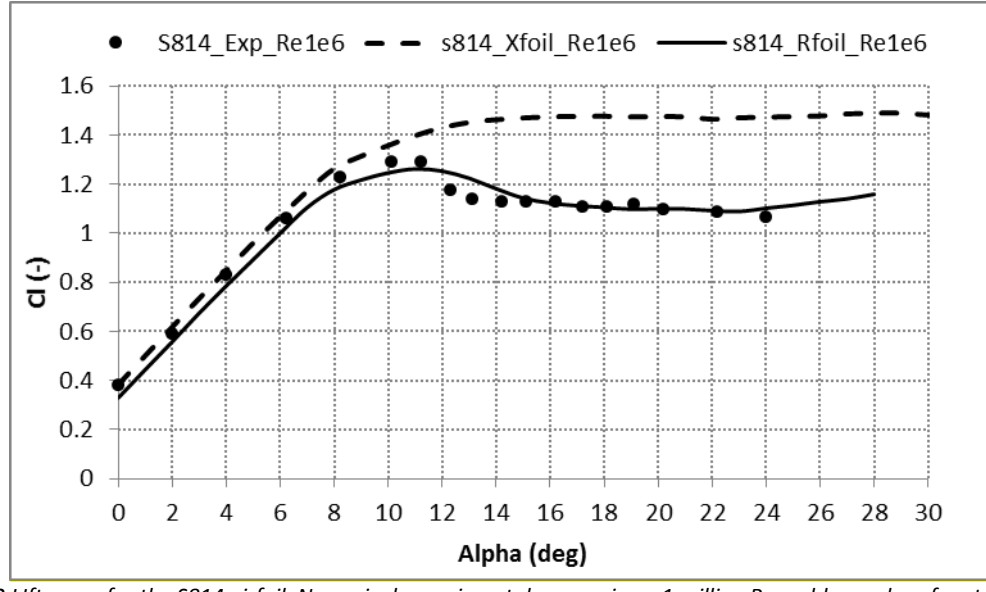

Figure 2 LIft curve for the S814 airfoil. Numerical experimental comparison. 1 million Reynolds number, free transition.

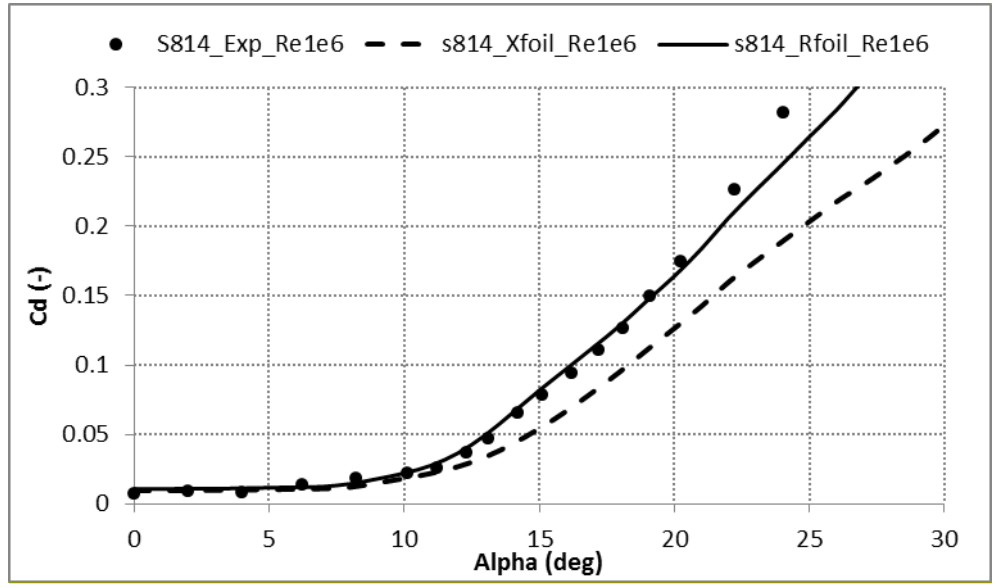

Figure 3 Drag curve for the S814 airfoil. Numerical experimental comparison. 1 million Reynolds number, free transition.

The geometry of the airfoil is parameterized (Grasso, 2008) with a combination of four Bezier curves (see Prautzsch et al., 2002, Barsky, 1990, Beach, 1991 for general information about Bezier curves) of third order distributed along the airfoil contour (figure 4). Each Bezier curve covers one quarter of the shape with 13 control points free to move in chord and normal-to-the-chord directions (i.e. 26 design variables). To appreciate and understand the choice of four Bezier curves, the reader should consider that third order polynomial is needed to describe inflection points; however higher degree can lead to wavy shapes. Dividing the airfoil contour in four pieces is a smart move to divide the complexity of the parameterization and ease the control of the shape. This formulation is C2 continuous. 15 design variables are active in the present work; in fact, the leading edge cannot move, while the neighbours and the trailing edge can move only in vertical direction. In addition, the control points 4 and 10 are internally controlled to ensure C2 property also in those points. The complete mathematical formulation can be found in Grasso, 2008.


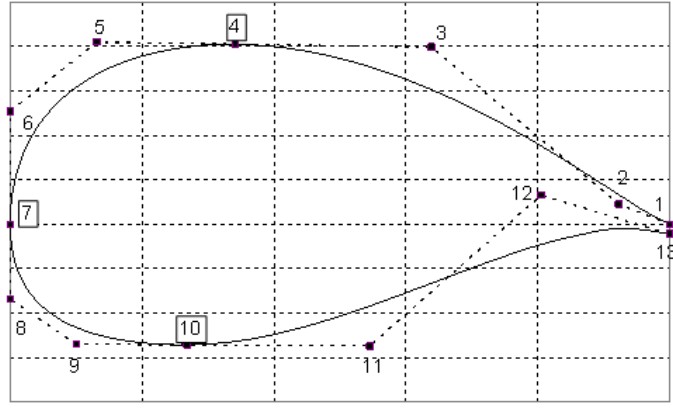

*Figure 4 Airfoil shape parameterization scheme. From Grasso, 2008.*

**3. Results**
*3.1. Airfoil performance*
The blade in development has only two airfoils (one main and one at the inner part, excluding the
blending area at the very root of the rotor) in order to simplify the blade construction. The first one is a
30% thickness airfoil which is used at the maximum chord station, while the second one is a 25%
thickness airfoil which extends from the half of the blade span to the tip. A blending area connects these
two airfoils. This work focuses on the main airfoil design where the main target is the aerodynamic
efficiency (L/D) maximization at the operative Re number of 1 million. At the same time, appropriate
stall behaviour needs to be achieved in order to provide good control to the wind turbine, while
minimizing stall induced vibrations.
As already mentioned, this aspect plays a crucial role in the present work. From optimization point of
view, several options in terms of constraints and design points to be included are possible. Some of them
are discussed here. High lift performance may lead to sharp stall behaviour, a constraint limiting the
maximum lift coefficient can be quite natural choice. However, limiting the lift coefficient at a specific
angle of attack may not be sufficient since there will be no control on different angles. The risk would
then be that the stall angle could simply delay or anticipate, making the constraint (technically satisfied)
completely ineffective. The same constraint could be then assigned simultaneously for several angles of
attack around the expected stall angle range. This will gain little more confidence but it will add
complexity to the optimization problem and increase the computational costs. Even more dangerous, the
risk of limiting too much the design space and drive the solution to local optima would increase. Anyway
there will be still no guarantee about post-stall characteristics, which would still require specific
constraint(s). A better and more accurate approach could be evaluating the full polar at each design
iteration and retrieve the information about maximum lift coefficient and post-stall (via for instance the
lift slope value). In this way, the number of constraints will reduce to just two which would fully describe
the stall behaviour, while keeping low the mathematical complexity of the optimization problem.
However, the computational time would rise because the full polar needs to be calculated for any
iteration. On top of that, the same approach should be used in rough conditions to make sure that the
airfoil has comparable characteristics in both cases.
Although the latest approach would be the most accurate, a different and more practical solution has
been adopted in the present work, which should have still good level of accuracy. A combination of
constraints focused on maximum lift coefficient (<1.4) and moment coefficient (> -0.12) has been
prescribed. In fact both constraints act on the shape of the lift curve bounding its maximum point and
its average position in lift axis (i.e. defining the alfa zero lift or the lift at zero degrees), respectively.
Considering the airfoil geometry, both constraints have a direct impact on the camber line of the airfoil
and their combined effect is to get soft stall with no excessive cambered shape. Since the roughness
generally has little influence on the linear region of the moment coefficient curve, the same constraint
on clean conditions should cover also the rough condition.
The airfoil thickness (t/c) of 0.25 has been selected, rather than a thinner value. Although the pure
aerodynamic performance could be better with thinner (e.g. t/c 0.15, 0.18) airfoils, thicker sections offer
the advantages of saving blade mass and provide higher strength to the blade structure.
Considering existing airfoils, the S821 and the S819 have been used as reference (Somers, 1993, Tangler
et al., 1995, Somers, 1998) because of their good characteristics in terms of insensitivity to roughness
and post stall behaviour. Figure 5 shows the shapes, while figures 6 – 8 show the aerodynamic
performance of these airfoils in free and fixed transition, as calculated with the RFOIL code. The
Reynolds number used for the simulations is 1 million, in accordance with the average real Reynolds
number value expected for a 60kW-range machine. All the simulations in fixed transitions
(representative of rough condition) presented in this work prescribe transition at 5% of the chord on the
suction side and 10% of the chord on the pressure side. It should be noticed the stall and post-stall
behaviour that is soft but monotonically decreasing in the indicated angle of attack range. In addition, it
should be noticed the relative small margin between the design point and the stall; for stall-regulated
turbines, this is an important feature to avoid excessive loads once the design condition has been passed
(e.g. in case of wind gust).

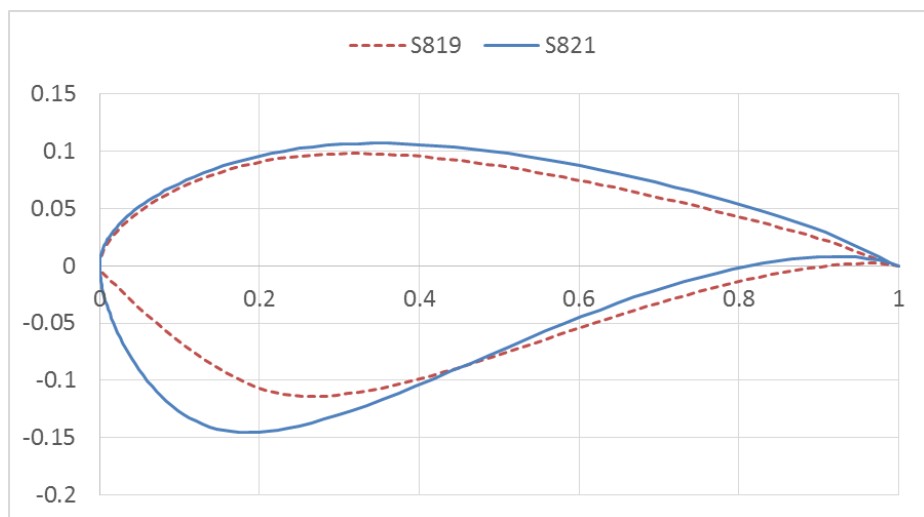

*Figure 5 S819 and S821 shapes.*

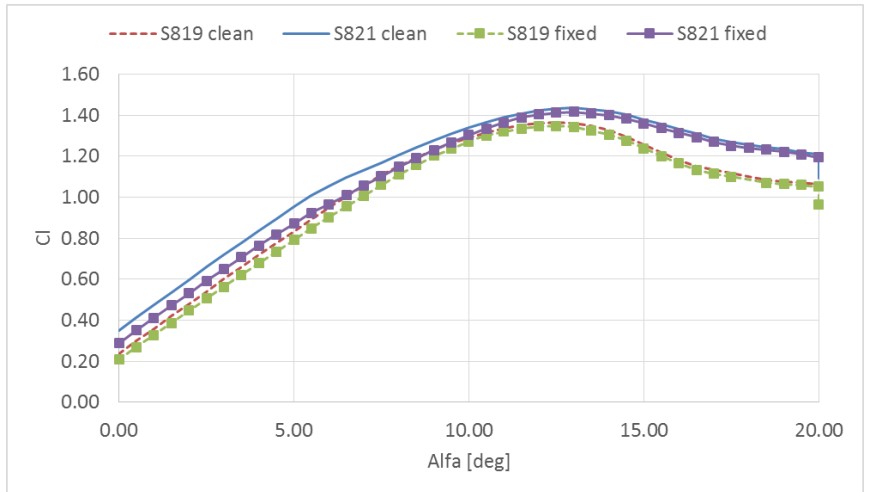

*Figure 6 Lift curves for S819 and S821 airfoils. Free and fixed transition data, 1 million Re number. RFOIL predictions.*

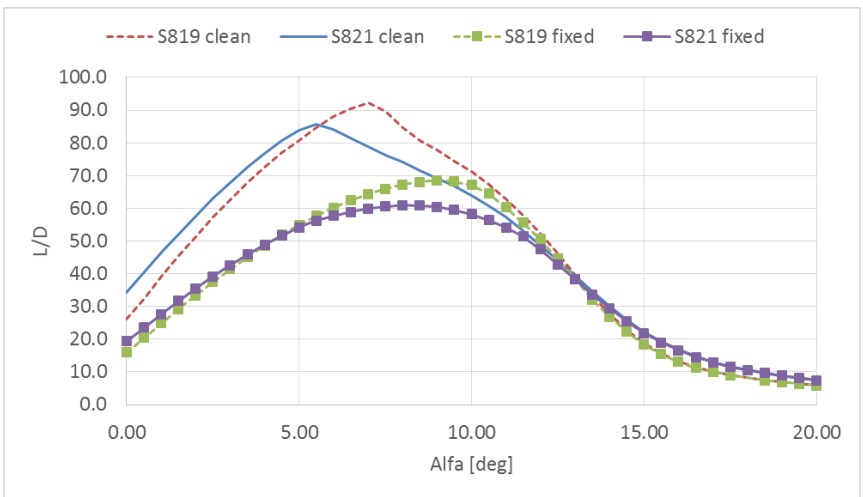

*Figure 7 Aerodynamic efficiency curves for S819 and S821 airfoils. Free and fixed transition data, 1 million Re number. RFOIL*
*predictions.*

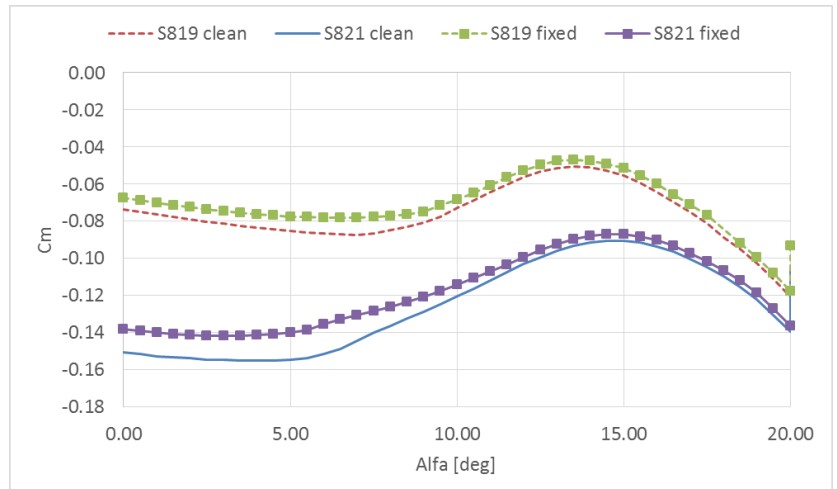

*Figure 8 Moment coefficient for S819 and S821 airfoils. Free and fixed transition data, 1 million Re number. RFOIL*
*predictions.*
So the ideal airfoil is a 25% thick shape (similar to the S821 which is 24% thick) with L/D performance
similar to S819, reduced stall margin and maximum lift coefficient (Clmax), but also small roughness
sensitivity and contained moment coefficient (Cm); the latter to avoid excessive torsional loads.
With these parameters in mind, three airfoils have been developed to offer better performance than the
reference geometries. The airfoils have been preliminary named A, B and C and are all 25% thick (the
shapes are not shown because of confidentiality issues). Their aerodynamic characteristics, evaluated
with RFOIL, are illustrated in figure 9 and 10.
The airfoil A has more camber than the other airfoils since the constraint on moment coefficient
discussed above has not been used in order to check the validity of the assumption. This is evident from
the lift curve. It achieves better efficiency in clean condition. However, its behaviour is very sensitive
to the roughness; in fixed transition the efficiency drops significantly and the lift curve changes
completely, making the control of the wind turbine impossible. The differences are smaller for the airfoil
B, but the post-stall characteristics of the lift curve make the control of the turbine difficult. The airfoil
C (from now on, called G25sx6) is instead a good compromise between good performance and good
control properties. The lift curve is in practice almost unchanged from free to fixed transition, as result
of adopting the constraint on moment coefficient and lift coefficient. In addition, the stall angle of attack
is unchanged. In terms of efficiency, the G25sx6 exhibits the best performance in fixed transition and a
quite flat plateau in both free and fixed transition. As mentioned, this is quite convenient for stall
regulated turbines because the airfoil will operate in a range of angles of attack rather than a specific
value like in the pitch controlled machines. Combining lift and efficiency performance, the stall margin
is almost unchanged between free and fixed transition.

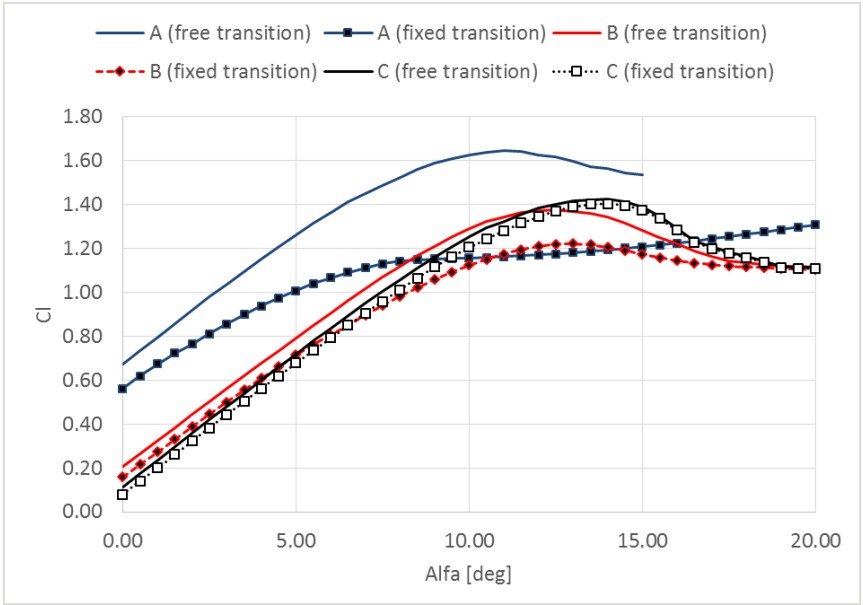

*Figure 9 Lift curve of the new airfoils. Free and fixed transition data, 1 million Re number. RFOIL predictions.*

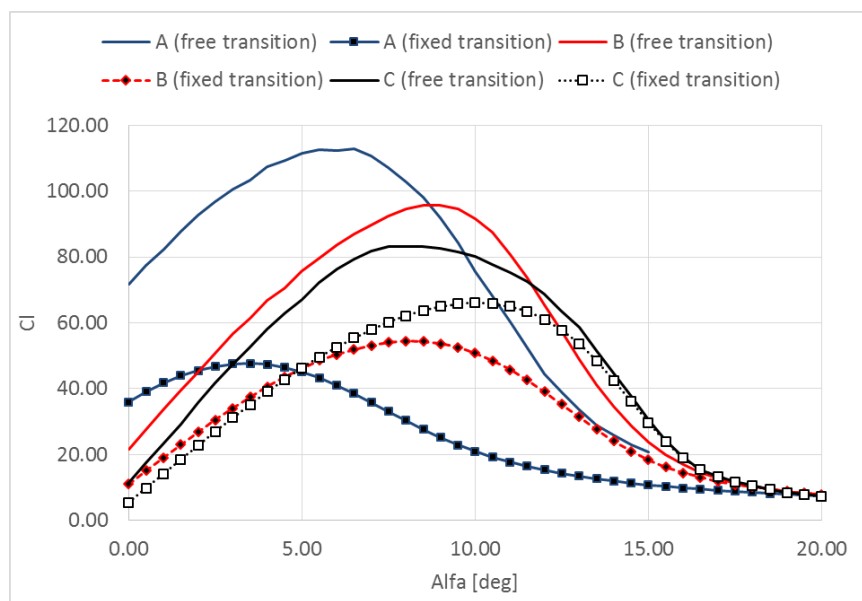

*Figure 10 Aerodynamic efficiency curve of the new airfoils. Free and fixed transition data, 1 million Re number. RFOIL*
*predictions.*
Comparing the G25sx6 with the S821 airfoil (figures 11 and 12) a similar value of efficiency in free
transition can be noticed but better performance in fixed transition despite the G25sx6 being thicker
(25%) than the S821 (24%).
In addition, the efficiency curves keep a good level over a wider range of angles of attack and the stall
margin is reduced, that is an advantage for stall regulated wind turbines (i.e. avoiding excessive loads
in case of wind gust).

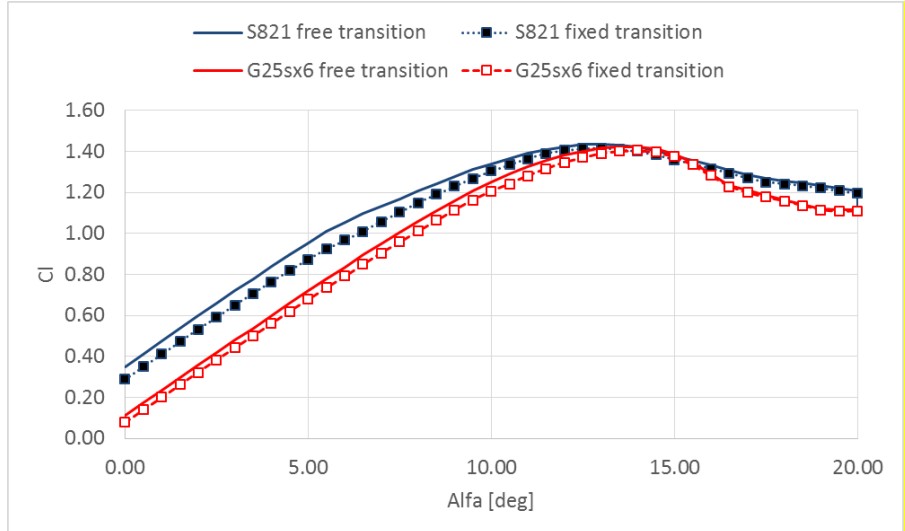

*Figure 11 Lift curve of the new airfoil. Free and fixed transition data, 1 million Re number. RFOIL predictions.*

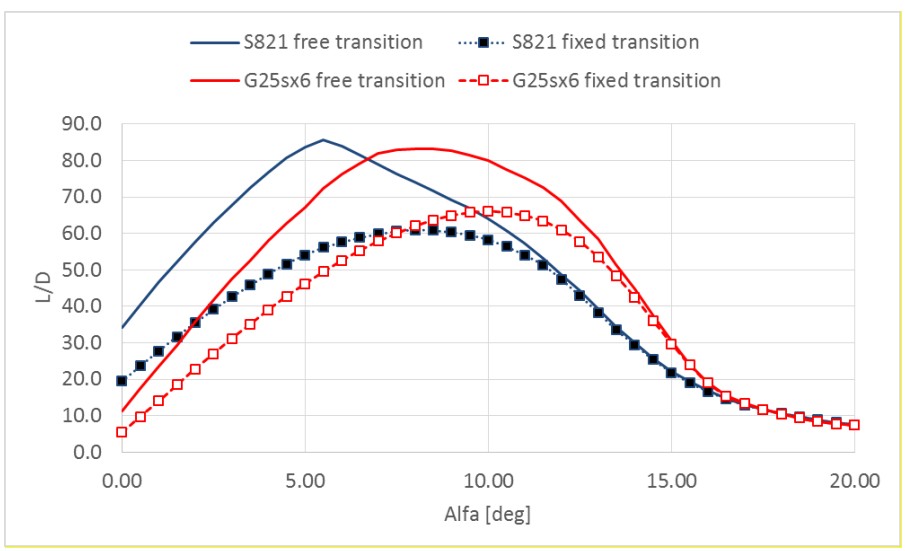

*Figure 12 Aerodynamic efficiency curve of the new airfoil. Free and fixed transition data, 1 million Re number. RFOIL*
*predictions.*
*3.2. Optimization process details*
This section presents some of the details of the optimization process for the G25sx6 airfoil. As
mentioned in the previous paragraph, the L/D was used as parameter to be maximized. To obtain good
roughness robustness, the design has been performed in fixed transition conditions; in addition, the L/D
value was divided by a factor 10 to have the same order of magnitude (o1) used for the constraints.
Figure 13 shows the evolution of the objective function during the iterations of the optimization process.
As it can be observed, the trend is not monotonically increasing as one could expect. This is because, to
reduce the risk to obtain a local optimal solution, the NACA0012 airfoil has been used as initial solution,
which is out of the feasible domain (t/c violating the threshold value) and so far from any possible
feasible local optima. The optimization algorithm is designed to obtain first a feasible solution (if any)
and then optimize it inside the domain space. Roughly the first 100 iterations are used to obtain a feasible
solution. This is evident by looking at figure 14 where the evolution of the constraints is illustrated,
together with their threshold values identified by the division between the feasible domain (blue area)
and the unfeasible one (red area). The circle in figure 13 corresponds to the optimal solution.

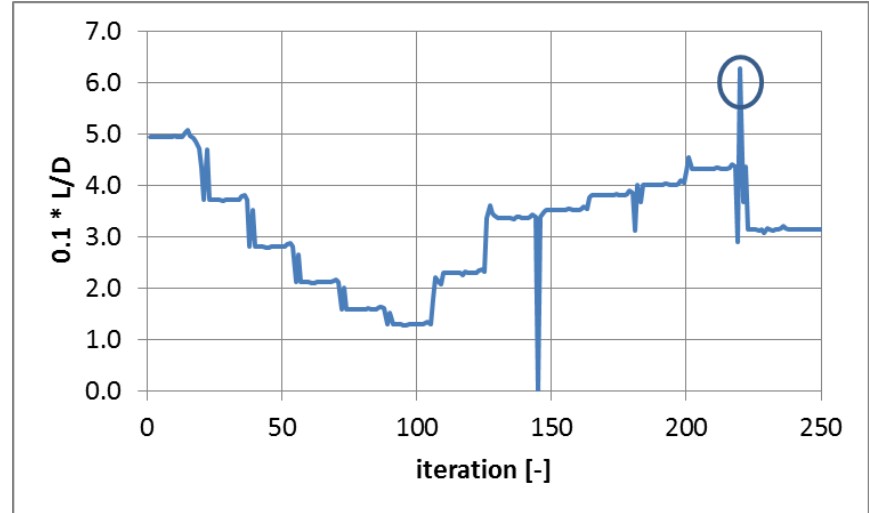

*Figure 13 Evolution of the objective function during the design process. Optimal solution highlighted in the circle.*

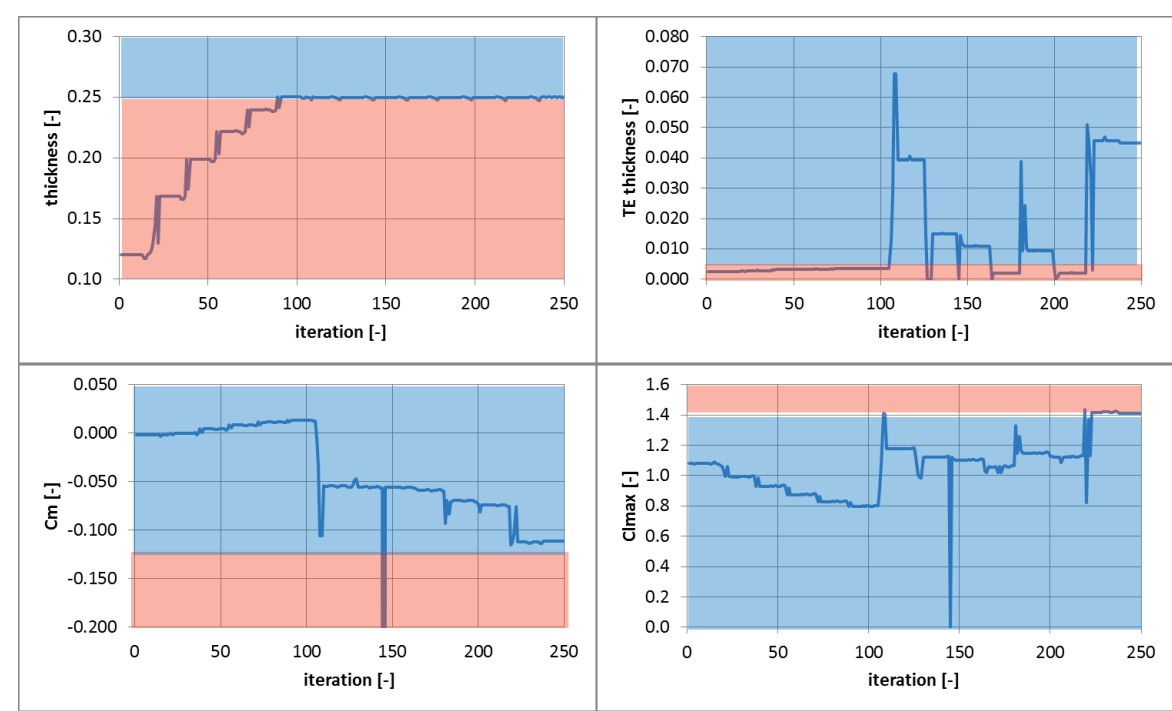


*Figure 14 Evolution of the constraints during the design process. The blue region corresponds to the feasible domain, while*
*the red one corresponds to the unfeasible domain.*
*3.3.  Impact on rotor performance*
In order to assess the value of the new airfoil, its impact on wind turbine performance has been evaluated
with a numerical analysis. This step is important to give a complete overview of the new airfoil effects,
but is actually necessary to make sure that the optimization problem has been correctly setup and the
constraints are effective in preventing or limiting stall-induced vibration since only the airfoil side of
the problem has been investigated after being separated from the rotor response.
A 60kW stall-regulated wind turbine has been used as reference and the S821 and G25sx6 airfoils have
been adopted as main airfoil. The reference wind turbine is a three blades machine designed to produce
energy in sites characterized by a very low mean wind speed (4m/s). Thus, its main characteristics are
very low values of cut-in and peak power wind speeds (about 2.5 m/s and 8.5 m/s respectively) and a
high AEP with a mean wind speed of about 4 m/s. To obtain this performance a generous rotor radius
and particularly slender blades are adopted: the radius is 14 m and the rotational speed is constant 34
rpm.
Figure 16 shows the power curves for the blade optimized based on the S821 airfoil and G25sx6 airfoil.
The BEM-based (Hansen, 2007) tool WT_Perf (Buhl, 2004) developed by the NREL has been used for
these analyses.
The blade geometry has been adjusted to consider the actual airfoils adopted. Normally, this includes
chord and twist; however in this case, the same chord distribution has been used (figure 15) since
preliminary analyses showed little impact on overall performance.

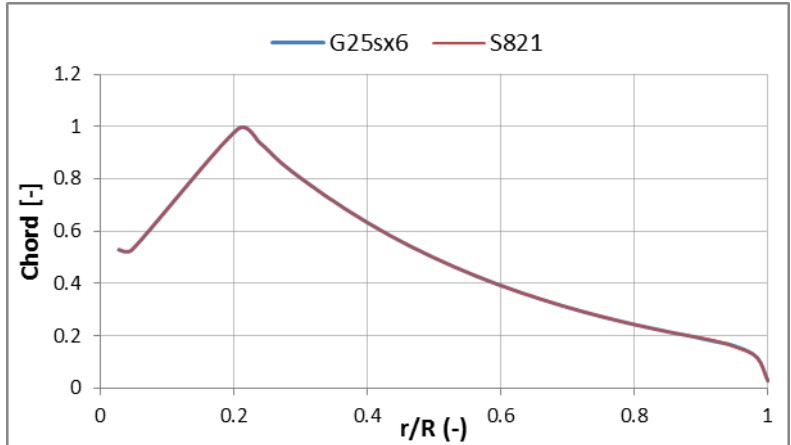

*Figure 15 Chord distribution adopted during the blade design.*
As already mentioned, the G25sx6 is 1% thicker than the S821; this ensures a higher moment of inertia
of each section implying a lower weight of the blade. From a preliminary analysis, the weight of the
blade can be reduced of about 5%.
Both free and fixed transition conditions have been included, as representative of clean and rough blade
conditions. According to the results, there are no symptoms of stall-induced vibration. This was not
expected anyway to happen but since the airfoil design has been performed in fixed transition, the
performance in clean condition could have been subject to risk of stall-induced vibration.
The power curves related to free and fixed transition in the figure refer to different values of the blade
pitch, which is the value necessary to achieve the desired peak power in each case.
Since in fixed transition the lift coefficient (particularly the maximum lift coefficient) is lower than in
free transition, a larger value of pitch angle will be necessary to reach the desired peak power. At the
same time, higher wind speed is needed to reach the same peak power.

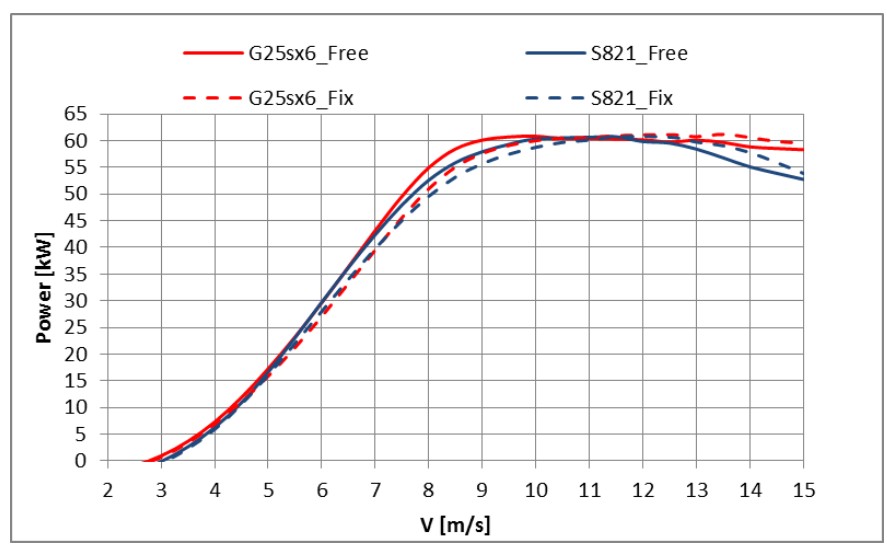

349                        *Figure 16 Effect of the new airfoil on the wind turbine power curve.*

The following figure shows the angle of attack distribution along the blade at 5 m/s and in free transition
condition for both the wind turbines. The unusual distribution that can be noticed at the tip of the blade
is due to the twist distribution adopted to reduce stall-induced vibrations, reduce the loads and improve
the overall stability; this feature, together with the rest of the blade design strategy and process will be
discussed in a dedicated work.

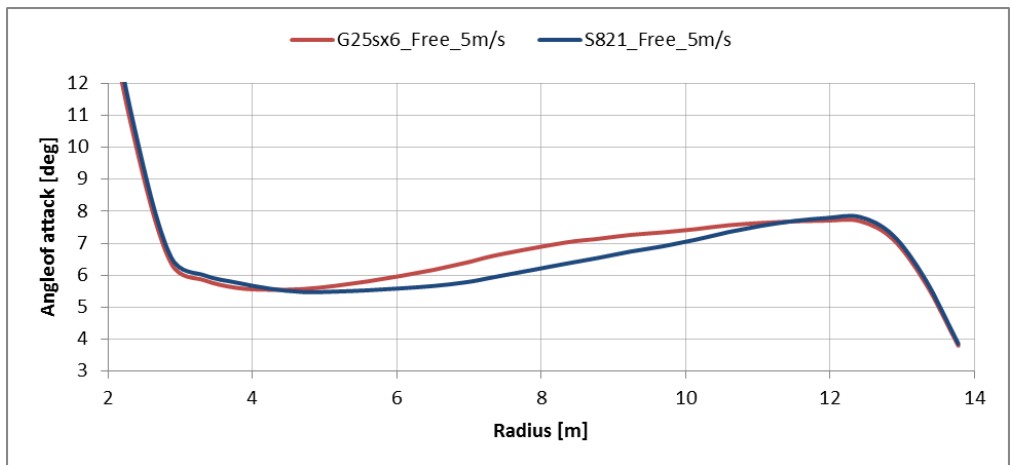

357                        *Figure 17 Angle of attack distribution along the blade.*

358                        *Table 1 Impact of the new airfoil on the wind turbine AEP.*

| Airfoil | Free transition | | Fixed transition | |
|---|---|---|---|---|
| | AEP [kWh] | Δ [%] | AEP [kWh] | Δ [%] |
| S821 | 136000 | - | 129000 | - |
| G25sx6 | 143000 | +5.15 | 132000 | +2.3 |

Considering the overall Annual Energy Production (AEP, see table 1), the new airfoil provides a
considerable gain in free (+5.1%) and fixed (+2.3%) conditions. More in detail, the turbine reaches the
maximum power for lower wind speed and the post-peak region is smoother. In addition, the production
at very low wind speed increases thanks to the new airfoils.

*3.4. Verification of WT_Perf for rotational effects*
The findings illustrated so far are based on BEM assumptions and WT_Perf accuracy. In particular, the
flow at the root is a critical point. In fact, lift and drag coefficients of root airfoils of a rotating blade are
affected by the so called 'stall delay' phenomenon (Himmelskamp, 1947, Guntur,2011, Herráez, 2014);
so the two dimensional aerodynamic curves of these airfoils need to be adjusted at high angles of attack
before being used in a BEM code like WtPerf to consider rotational effects. In this work, lift and drag
coefficients of the inner airfoils (approximately from root to 20% of the blade) have been extrapolated
from a CFD analysis of a rotating blade following the inverse BEM method reported in Guntur, 2014,
while two-dimensional aerodynamic coefficients obtained by using RFOIL have been used for the
airfoils along the outer half of the blade, where rotational effects can be neglected (Tangler, 2005). This
method is useful to speed-up the wind turbine optimization process because it allows to modify the outer
part of the blade, which is most influential for the performances and behaviour of the whole system,
simply using two-dimensional aerodynamic airfoil characteristics.
One of the preliminary blades designed during this work has been used as reference to validate the
method. Despite the design was intended to produce a 60kW machine, the actual results ended in a
rejected design since it failed to be controllable. This fact however, made the design an interesting test
case for validation because of two distinct peaks in the power curve. Figure 18 shows the comparison
between the power curve predicted with CFD analysis in steady operating conditions and the power
curve obtained with the method used in this work. STAR-CCM software has been used, with the k-ω
turbulence model. As it can be noticed, the agreement is very good, as the BEM-based scheme captures
not only the general trend but also the two peaks at 12m/s and 15m/s wind speed. A publication dedicated
to the topic is under preparation at the moment which will provide the full details on the development
done on the subject.

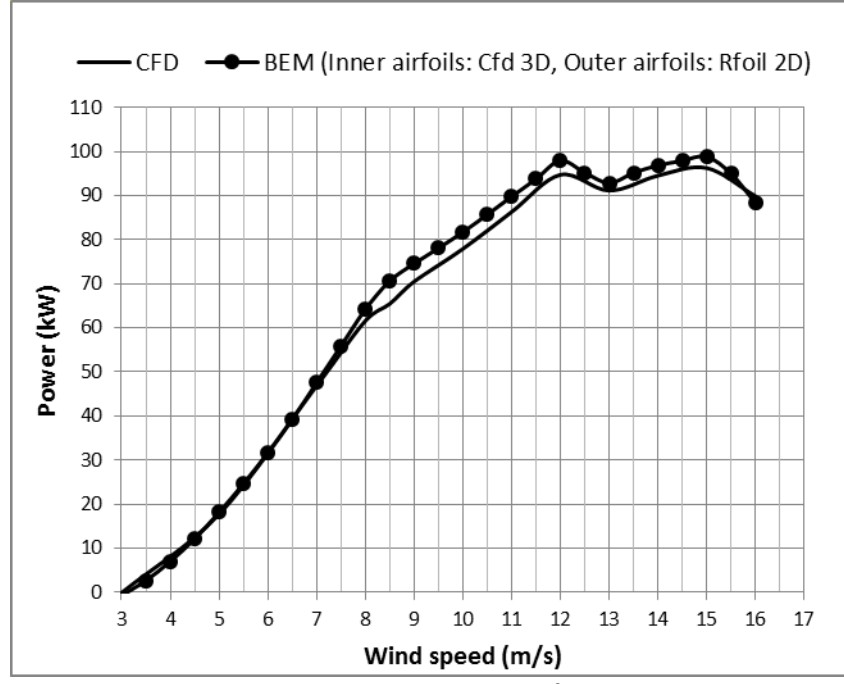

*Figure 18: CFD power curve VS Wt_Perf power curve*
*implementing aerodynamic curves of inner airfoils extracted from CFD*
**4. Conclusions**
Despite the pitch controlled wind turbines cover the complete large MW machines market, stall
regulated solutions are still diffused for small power production. A new airfoil specifically designed for
this class of wind turbines has been developed and presented in this work. Compared to existing

geometries, the new airfoil can increase the annual energy production of the machine, both in clean and rough conditions. In terms of rotor performance, the new airfoil brings an evident benefit on the punctual power production and on the overall AEP (+5.1% in free transition and +2.3% in fixed transition).

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
