# Peer review of "Design of advanced airfoil for stall-regulated wind turbines"

_Wind Energy Science, 2017_

## Referee Comment (RC1) · Anonymous Referee #1 · 17 Mar 2017

**Wes 2017-5 Design of advanced airfoils for stall-regulated wind turbines - Grasso et al.**

The paper treats the design of airfoils for small stall controlled wind turbines using an optimizer, focusing on induced-vibration sensitivity and peak power.

The following remarks can be made:

- The authors are advised to check the English in their paper, since there are many errors

About the design methodology

The value of the lift and drag in the stalled region are of primary importance for the prediction of power and of the induced vibration sensitivity of a stall controlled turbine. RFOIL is used to calculate the airfoil performance in the optimization process. The authors give no evidence that RFOIL is predicting the characteristics well for the stalled region. I would expect at least a couple of comparisons with wind tunnel experiments to show how well RFOIL is doing in this respect. In short: what is the predictive value of RFOIL for this class of airfoils (thickness, maximum lift) and Reynolds numbers?

Furthermore it is stated that WtPerf is used. What is the input to this program for the high angles of attack to calculate the power? What method was used? In what part of the power curve (e.g. figure 1) can we see the differences in airfoil post stall lift and drag ?

Figure 1 shows the lift curves for 2 airfoils. Without additional information it is not clear how this lift performance contributes to the power curves at the right hand side.

Page 5, lines 183-190 about the maximum lift coefficient. Please rephrase. Now it is not so clear what is meant here.

Figures 4 to 10: Please use one color per airfoil and e.g. a solid line for the clean case and one with symbols for the rough case.

Page 7, line 226. What exactly is the design point for this type of airfoils?

Page 10, line 295.  Generally a coastal region is not associated with a low wind speed.

Figure 14 is quite misleading actually. In practice the pitch is constant for a stall turbine. The effect of fixed transition is lower power.

Talking about fixing transition: at what chord location was transition fixed?

Table 1: what wind regime was used to calculate the AEP?

Conclusion: this is a purely theoretical study. Without any reference to reality it remains unclear if this airfoil indeed leads to the predicted increase in rotor performance.

---

## Author Comment (AC1) · 23 Mar 2017

Dear Editor and Reviewer(s),

Thanks for your comments. In the text below, the reaction from the Author and co-Authors is added in red for each comment.

**Wes 2017-5 Design of advanced airfoils for stall-regulated wind turbines - Grasso et al.**

The paper treats the design of airfoils for small stall controlled wind turbines using an optimizer, focusing on induced-vibration sensitivity and peak power.

The following remarks can be made:

- The authors are advised to check the English in their paper, since there are many errors

About the design methodology

The value of the lift and drag in the stalled region are of primary importance for the prediction of power and of the induced vibration sensitivity of a stall controlled turbine. RFOIL is used to calculate the airfoil performance in the optimization process. The authors give no evidence that RFOIL is predicting the characteristics well for the stalled region. I would expect at least a couple of comparisons with wind tunnel experiments to show how well RFOIL is doing in this respect. In short: what is the predictive value of RFOIL for this class of airfoils (thickness, maximum lift) and Reynolds numbers?

It is true that such comparisons are not in the paper, but they can be found in the references. In particular in van Rooj 1996 and Grasso 2011. However the graphs below will be added in the next review.

[Figure]

[Figure]

Furthermore it is stated that WtPerf is used. What is the input to this program for the high angles of attack to calculate the power? What method was used? In what part of the power curve (e.g. figure 1) can we see the differences in airfoil post stall lift and drag ?

As clearly written in the text (line 304), Wtperf is based on BEM. It has been considered very well known what BEM is and what the assumptions are. So the theoretical and implementation details have been considered redundant. Those can be found in the references (also in line 304).

About the power curve, considering the stall-regulated nature of the machine under investigation, this implies that the pitch angle of the blade is fixed. This means that the operative sectional angle of attack increases with the wind speed, which leads to the fact that passed the rated power wind speed, the power decreases because the aerofoils enter in the stall region. This is actually used to control the machine, as explained in section 2.2. General theory can be found in the referenced Hansen 2007.

Figure 1 shows the lift curves for 2 airfoils. Without additional information it is not clear how this lift performance contributes to the power curves at the right hand side.

See previous point

Page 5, lines 183-190 about the maximum lift coefficient. Please rephrase. Now it is not so clear what is meant here.

Text replaced

Figures 4 to 10: Please use one color per airfoil and e.g. a solid line for the clean case and one with symbols for the rough case.

Figures 4-6 are already in line with the indications. Proper adjustments will be done to the other figures. However, it is not clear if the figures should be b/w or can be coloured.

Page 7, line 226. What exactly is the design point for this type of airfoils?

As explained in section 2.1 and 2.2, the design point (either in terms of lift coefficient or angle of attack) should be anyway such to guarantees the maximum L/D performance. The main difference with the airfoils used on pitch-regulated machines is that the design point should be close to maximum lift value.

Page 10, line 295. Generally a coastal region is not associated with a low wind speed.

I guess it may depend on the specific country. In Italy the areas with strongest wind are inland along the Appennini, while coastal areas have 4m/s average wind speed. The work focuses on low wind sites despite their location. To avoid confusion, the sentence can be rephrased.

Figure 14 is quite misleading actually. In practice the pitch is constant for a stall turbine. The effect of fixed transition is lower power.

Correct, fixed transition is indicative of rough conditions and so lower power. However, the figure shows HOW the power curve is changing in consequence of rough conditions. In particular, it shows how the power decreases after nominal power. It can be seen that the new airfoils preserve a flat power curve also in rough condition.

Talking about fixing transition: at what chord location was transition fixed?

All the data in fixed transition are calculated assuming transition at 5% on the suction side and at 10% on the pressure side. Somehow this information has been omitted. It will be added.

Table 1: what wind regime was used to calculate the AEP?

It was assumed an average wind speed of 4m/s as representative of low wind sites. This is consistent with the cut-in and cut-out velocities mentioned in the paper. Strictly speaking however, for the sake of a fair comparison, which is the goal of the work, the actual wind regime is not relevant provided that it is the same for both designs.

Conclusion: this is a purely theoretical study. Without any reference to reality it remains unclear if this airfoil indeed leads to the predicted increase in rotor performance.

Quite disagreed: the study presented is necessarily a numerical investigation since the real wins turbine is under-construction. However, given the general reliability of BEM method (widely used in the Industry) and most importantly, the accuracy of RFOIL code, it is opinion of the Author and co-Authors that the reality will reflect the results described in the present work. Especially in terms of relative variations.

---

## Referee Comment (RC2) · Anonymous Referee #2 · 27 Mar 2017

The paper address a relevant scientific questions within the scope of WES. Although not "revolutionary", the paper presents an original mix of design tool and procedures. Thus the paper is, in my opinion, of broad international interest. The Objectives are very clearly outlined. Considering the hypotheses, I would appreciate a discussion about the "2D stall assumption" implicitly contained in this design procedure. Under the mentioned 2D assumption, the methods are valid and are indeed very well described. Thus I agree with reviewer N.1 that the reliability of RFOIL stall prediction must in some way documented but, furthermore, I think that a sort of validation of the whole rotor behavior (including 3D effects on the stall) should be in some way included (or, at list, this point should be discussed). An interesting article including this point is: A. Le Pape* and J. Lecanu, 3D Navier–Stokes Computations of a Stall-regulated Wind Turbine, Wind Energ. 2004; 7:309–324 (DOI: 10.1002/we.129). Beside this "weakness" the article is very clear, and well written. As a last point, I would suggest to include a more complete list of references so that the article results more framed in the existing literature.

---

## Author Comment (AC2) · 9 Apr 2017

Dear Reviewer,

thanks for your comments.

Concerning RFOIL accuracy, as already mentioned in our reaction to reviewer1, the references contain already some validation of the code. However, more details can be added in the present work to make more clear the point.

About general accuracy of BEM tool for design of stall regulated machines, the calculations done to obtain the presented data have been verified with 3D CFD analysis, where the flow at the root region of the blade

best regards

Francesco Grasso on behalf of the authors

---

## Referee Report (RR1)

**Wes 2017-5 Design of advanced airfoils  for stall-regulated wind turbines - Grasso et al.**

**Comments on the revised manuscript.**

Although quite some errors have been removed, still the English used could be better. Below a number of suggestions to improve the text.

Page 1 line 7 at the end: _In_ the design phase

Line 12: add _the_ between with and RFOIL

Line 31: exchange the comma with _and,_ move _correctly_ to the end of the sentence

Page 2 line68: add _was_ behind _what_

Page 3 line 102: _intended?,_ probably here it is _regarded_

Line 130: move  _an airfoil……stall_ to behind _using_

Page 4 line144: _introduce_s , Rephrase the rest of the sentence

Page 6 line 217: same = comparable

Page 8 line264: move _impossible_ to the end of the sentence. Do the same with _difficult_ in the next sentence

Line 263: roughness=fixed transition (or: transition close to the leading edge), remove the comma behind transition

Page 9 line 280: remove _it_ and move _can be noticed_ to behind _transition_

Line 281: is=being

Document: do not use _power peak_ but rather use _maximum power_ or _peak power_

Page 15: remove the words _visible_ and _visibly_

---

## Author Response (AR2)

Dear Editor and Reviewer(s),

Thanks for your comments.

All the corrections about English language in the text have been corrected by following the reviewer's suggestions.

Best regards

Francesco Grasso